# A Comparative Study of Lymphatic Filariasis-Related Perceptions among Treated and Non-Treated Individuals in the Ahanta West Municipality of Ghana

**DOI:** 10.3390/tropicalmed7100273

**Published:** 2022-09-28

**Authors:** Collins Stephen Ahorlu, Joseph Otchere, Kojo M. Sedzro, Sellase Pi-Bansa, Kofi Asemanyi-Mensah, Joseph L. Opare, Bright Alomatu, Elizabeth F. Long, Dziedzom K. de Souza

**Affiliations:** 1Epidemiology Department, Noguchi Memorial Institute for Medical Research, College of Health Sciences, University of Ghana, Legon, Accra P.O. Box LG 581, Ghana; 2Parasitology Department, Noguchi Memorial Institute for Medical Research, College of Health Sciences, University of Ghana, Legon, Accra P.O. Box LG 581, Ghana; 3Neglected Tropical Diseases Programme, Ghana Health Service, Accra P.O. Box MB-190, Ghana; 4Neglected Tropical Diseases Support Center, Task Force for Global Health, Inc., 330 West Ponce de Leon Ave, Decatur, GA 30030, USA

**Keywords:** lymphatic filariasis, MDA, non-treated, treated, engage and treat, test and treat, Ghana, Ahanta West, perceptions

## Abstract

**Background:** Ghana joined the Global Programme to Eliminate Lymphatic Filariasis (GPELF), established in the year 2000, with the aim of eliminating the disease as a public health problem through annual mass treatment of entire endemic populations. Since 2001, the country has implemented mass drug administration (MDA) in endemic districts, with great reductions in the population at risk for infection. However, in many districts, the elimination programme is faced with the presence of hotspots, which may be due in part to individuals not taking part in MDA (either intentionally or unintentionally) who may serve as reservoirs to sustain transmission. This paper compares the LF-related perceptions among individuals who regularly take the MDA drugs and those who seldom or never take part in the MDA in the Ahanta West Municipality of Ghana to determine community acceptable ways to implement an intervention aimed to track, engage, and treat individuals who regularly miss MDA or to test individuals who intentionally refuse MDA and treat them if positive for LF. **Methods:** This was a mixed method study employing questionnaire surveys and focus group discussions (FDG) for data collection. Survey participants were randomly selected from the 2019 treatment register to stratify respondents into treated and non-treated groups. FGD participants were selected purposively such that there are at least two non-treated persons in each discussion session. **Results:** Over 90% of the respondents were aware of the disease. Poor hygiene/dirty environment was wrongly reported by most respondents (76.8%) as the causes. MDA awareness was very high among both treated (96.9%) and non-treated (98.6%) groups. A low sense of vulnerability to LF infection was evident by a reduction in the number of people presenting clinical manifestations of the disease in communities. Slightly more, 65 (29.0%) of the non-treated group compared to the 42 (19.4%) treated group reported ever experiencing adverse effects of the MDA drugs. Barriers to MDA uptake reported in both groups were poor planning and implementation of the MDA, lack of commitments on the part of drug distributors, and adverse drug reactions. About 51% of the non-treated group reported never taking the drugs even once in the last five years, while 61% among the treated group took the MDA drug consistently in the past five years. Respondents in both groups believed that, when engaged properly, most non-treated persons will accept to take the drug but insisted community drug distributors (CDDs) must be trained to effectively engage people and have time for those they will be engaging in dialogue. The chiefs emerged as the most influential people who can influence people to take MDA drugs. **Conclusions:** The reduction in risk perception among respondents, adverse reactions and the timing of MDA activities may be influencing MDA non-participation in the study area; however, respondents think that non-treated individuals will accept the interventions when engaged properly by the CDDs.

## 1. Background

Lymphatic filariasis (LF), which is commonly referred to as elephantiasis, is a debilitating neglected tropical disease which leads to disability, thereby affecting the ability to work, reduced access to services and social stigmatization of affected individuals and families [1,2]. The clinical manifestations of the infection may also have important mental health consequences for the affected individuals, especially when they cannot receive the appropriate management of morbidity and disabilities associated with the disease which all together accounted for 5.09 million disability-adjusted life years [2,3]. Most infected persons are asymptomatic, but, when left untreated over several years, the infection could result in lymphedema of the limbs in both males and females, and hydrocele in males [4,5,6,7]. These clinical manifestations carry grave debilitating social and economic consequences for the infected person and his/her family [6,8,9,10]. LF is considered as the second most disabling condition, after mental illness globally [2,11,12].

The Global Programme to Eliminate Lymphatic Filariasis (GPELF) was established in 2000 with the aim of eliminating the disease as a public health problem [1,12]. The elimination of LF is targeted through: (i) the treatment of entire endemic populations through annual mass drug administration (MDA) with at least 65% treatment coverage for at least five years to break transmission [2,13,14] and (ii) the implementation of morbidity management to alleviate the suffering of individuals affected by the disease [12,15,16]. Many countries including Ghana have implemented MDA to endemic populations and have attained significant reductions in the population at risk [1,2,12,13,14,15,16,17,18]. Thus, the GPELF is one of the most cost-effective public health interventions [2,5,19]. There are however many challenges facing the elimination programme. In Ghana, some districts, referred to as “hotspots”, have received more than 15 years of treatment without interrupting transmission. This may be due, among other factors, to non-treatment of some individuals who may continue to serve as reservoirs to sustain transmission in their communities [2,6,13,14,15,20]. Innovative solutions to this challenge are therefore needed.

In our previous studies, we relied on the MDA registers to assess the non-treated and identify individuals who were not reached during MDA [2]. We proposed and implemented an Engage and Treat (E&T) strategy for individuals who unintentionally (i.e., sick, travelled, pregnant, unaware) missed MDA and a Test and Treat (T&T) strategy for individuals who refused MDA for fear of adverse events or low risk perceptions [2,21]. To guide the implementation of these strategies, this study was undertaken to compare the views of treated and non-treated individuals to determine the barriers and facilitators of MDA participation, explore the characteristic differences between the two groups, determine acceptability and best approaches to implement the E&T and T&T strategies in the Ahanta West District of Ghana (21). In this study, we defined “non-treated” to include: 1. Those who missed three consecutive MDAS. 2. Those who missed any three out of five consecutive MDAs. 3. Those who had never taken MDA medicine all.

## 2. Methods

### 2.1. Study Sites

The study was conducted in six communities in the Ahanta West district of Ghana, an area with persistent transmission of LF. The area has received more than 16 rounds of MDA without interruption of transmission [2,5]. One community was randomly selected from each sub-district, as a fair representation the entire district. Each selected community has a health facility.

### 2.2. Community Entry

Community entry was done by informing the regional, district and health facility officials of the Ghana Health Service (GHS) about the study objectives, methods, and timeline and their permission and support for the project was obtained. After that, the study team met with the officials of the sub-district health facilities, briefed them about the study and what is expected. The health facility officials then facilitated meetings between the research team and the community leaders. Data collection was done after gaining the support and willingness of community leaders to participate in the study.

### 2.3. Study Design and Data Collection

This was a mixed method study involving quantitative and qualitative data collection employing cross sectional questionnaire survey, focus group discussions (FGD), and in-depth interviews (IDI) as outline below. Data were collected in February 2020 to guide the implementation of the strategies targeting MDA defaulters immediately after the official MDA round. Informed consent was obtained from participants prior to their participation. Data were collected on the demographic characteristics of participants, disease awareness, causes of the disease, signs, and symptoms, means of control and prevention, mass drug administration, and adverse drug reactions.

### 2.4. Quantitative Data Collection

A cross sectional questionnaire survey was conducted with 440 respondents, stratified into treated (216) and non-treated (224) groups, using a structured questionnaire. Data were collected by means of Computer-Assisted Personal Interviews (CAPI) using the Open Data Kit (ODK) software. A server at Noguchi Memorial Institute for Medical Research (NMIMR) hosted the ODK application for the data collection, which was also used to store the data collected from the field. The selection of respondents was based on the treatment registers for the 2019 MDA, prior to the data collection. The treated group members were randomly selected from the MDA registers, while the non-treated group members were purposively targeted because there was no complete list available for non-treated group as there were many of them who were not captured in the treatment register, and we want to recruit some of them too into the study. A minimum sample size of 189 individuals was required for each group, bearing in mind that some of them may not be available to participate in the study. Quality control of the field survey performance during data collection was done using the distribution checks, survey duration, and GPS location audits. Quality of data, in terms of consistency and completeness, was checked every day by field supervisors.

### 2.5. Qualitative Data Collection

Trained and experienced qualitative data collectors conducted the FGDs and IDIs in the local Twi language. FGD data collection team was made up of a moderator, note taker, and a supervisor. The moderator followed a semi-structured discussion guide, developed around key thematic areas of interest. Each FGD session lasted approximately 50 to 75 min while an IDI session lasted between 30–40 min.

The discussions were audio-recorded and transcribed verbatim into English. Debriefing sessions were done after each day’s work to identify emerging themes that require further probing and ensure data quality on the field. A supervisor routinely reviewed all English translations and provided continuous feedback to the data collection team.

FGD participants were purposively selected adult community members, aged 18 years and above. To qualify for selection, a participant must have lived in the community for not less than 10 years and identified by community leaders as someone who is generally knowledgeable. An additional FGD session was conducted with the Community drug distributors (CDDs).

### 2.6. Data Analysis

The study’s data were analyzed with IBM SPSS Statistics version 20. Inconsistent responses were logically imputed after data were checked for inconsistencies. After visually analyzing the normality of these data with a histogram, the skewness and kurtosis tests were used to determine the normality of continuous variables. Frequency and percentages were used to describe categorical variables. The Chi square test of independence was used to determine the relationships between treated/non-treated and variables such as sex, age, and educational level. Assumptions were checked and cells that had expected count less then 5 were recoded. Statistical significance was set at a *p*-value < 0.05. Bonferroni tests were used to adjust the *p*-values for multiple-response variables.

Qualitative data were analyzed using an iterative process, beginning with the first interview, and continued throughout data collection through daily debriefings and review of data and emergent themes. The dialogues were transcribed directly into English and entered in a pre-coded format using Microsoft word, which was then imported into the MAXQDA software for qualitative data analysis. The MAXQDA software was used to further code relevant overlapping segments of the narratives and categorized statements for content analysis for the selection of representative and relevant responses for presentation. Both majority and minority views were presented. Coding was carried out using predetermined themes based on the research questions and key areas of interest [22]. A codebook was developed and refined as the analysis progressed. Coding of all transcripts was carried out by a trained research assistant and reviewed by the supervisor.

### 2.7. Ethical Approval

The study was reviewed by the Noguchi Memorial Institute for Medical Research (NMIMR IRB CPN 021/19-20). All participants provided written informed consent prior to participation.

## 3. Results

### 3.1. Characteristics of Respondents

A total of 440 respondents (216 treated and 224 non-treated) took part in the questionnaire surveys. More females (54.8%) were interviewed (57.9% treated and 51.8% non-treated). The majority of the respondents fall within the age group of 18 to 34 years (31.5% treated and 50% non-treated). A little over half of the respondents in both groups have had junior high school education or above (51.9% treated and 51.3% non-treated). Being a rural district, over 80% of the respondents in both groups were reportedly employed (89.8% treated and 86.4% non-treated), mostly in agriculture (farming and fishing) (Table 1).

Six focus group discussions (FGDs) were held (three each with male and female groups. In all, 42 participants (21 females and 21 males) took part in the FGD sessions (Table 2). A saturation point was reached after the fifth FGD, where no new data was coming out of the discussions.

### 3.2. Awareness/Knowledge of Elephantiasis

LF awareness was very high among respondents (96.9% treated and 98.6% non-treated). The main sources of information reported among others were knowing someone with the disease in the community (68.4% treated and 70.8% non-treated); the media, mainly television, radio, and newspapers (21.5% treated and 18.1% non-treated). Only a few mentioned community health workers as the source of their information on the disease (10.1% treated and 11.1% non-treated).

The poor hygiene/dirty environment was prominently and wrongly reported as a cause of the disease (72.7% treated group and 80.8% non-treated), alongside other different misconceptions. However, 15.7% and 11.6% of the respondents mentioned mosquito as the cause of the infection among the treated and non-treated groups, respectively (Table 3).

The high awareness of LF reported by the survey participants was confirmed by the qualitative findings, where respondents said they were aware of the disease, which they referred to as ‘*egyake dugba*’ in Ahanta and *‘**Gyepim*’ in Twi, the two dominant local languages spoken by the people in the district. These descriptions were largely due to the enlargement of the legs of people with the condition and the fact that it affects mobility of the affected individuals. The following quotes illustrate these points in support of the quantitative findings:

“It (LF) is a common disease here […]. We the Ahanta people do not call it ‘Gyepim’. We say the person has ‘Dugba’. The Fantes call it ‘Gyepim’ and the Ahanta people call it ‘Dugba’ […] when we want to make fun of the condition then we refer to it as ‘Egyake Dugba’”.(P6, FGD, Male treated, Bonsukrom)

“We know it (LF). We can see that one leg will be swollen/bigger than the other. And it will limit the person’s movements, that is why we called it Dugba”.(P2, FGD, Male non-treated, Egyambra)

### 3.3. Signs and Symptoms of LF

Various signs and symptoms of LF were reported. The most prominent signs/symptoms reported was ‘swollen legs’ (96.8% treated vs. 91.9% non-treated, Table 4).

Qualitative findings largely corroborate what was reported by survey participants about the recognition of signs and symptoms of LF. Some of the signs and symptoms mentioned were itching, swelling, big legs, as captured in in the following narratives:

“Before the disease will manifest, I was feeling heat and itching on my legs. Sometimes too, I feel pains in my groin and feverish. The big legs, as you see it now, took a long time to develop”.(P4, FGD, Male treated, Bonsukrom)

“The main thing that we know of is that the disease makes the legs of those infected big, usually one is bigger than the other. Some people too have their hand becoming bigger and we are told that it can also make you have *Etow* (hydrocele)”.(P5, FGD, Female non-treated, Achowa)

### 3.4. Treatment and Prevention

A majority of the survey respondents mentioned MDA as the main mode of treatment and prevention of LF in the study area. This was reported by 84.6% among the treated group and 76.5% among the non-treated group (Table 5).

Similar views were expressed during qualitative interviews; a majority of the participants were aware of the ongoing MDA in their communities. The MDA drugs, they claimed, serve as the main treatment for the disease and at the same time help to prevent people from getting the disease. This position was captured in the following representative narratives:

“We are all aware of the drugs that are given to us every year to treat the disease […] we all take it and whenever I asked, why should a person like me who is not having the disease should take the medicine? I was always told that it will protect me from getting the disease, So I have always taken it with my family […] it helps the children to release worms in their faeces and I think it is good”.(P3, FGD, Female treated, Mpatase)

### 3.5. Perceived Vulnerability Associated with LF and Infection

Participants generally admitted that LF is a common condition in the study area, especially in the past and they used to believe that it could affect anyone in the district at any time. However, they reported that the situation has changed over the years, especially since the introduction of the MDA interventions. Respondents in both treated and non-treated groups maintained that the number of people presenting clinical manifestations of the disease in their communities have reduced significantly, which they have attributed mainly to the implementation of MDA activities, and this has given them hope that they may not be at risk of infection anymore. These positions were captured in the representative narratives below:

“At first you can see a lot of people with the disease in most communities and at that time we used to think that the disease can affect all of us, but now you don’t see many people having it. […] I am sure it is because of the medicine being given to us that has helped to reduce the condition in our communities […]. Now, most of us think that we are no longer as risk of getting the diseases”.(P7, FGD, female treated, Mpatase)

“Although, I don’t always take the medicine for the disease, I believe that it is helping many people and by reducing the number of people with the disease in the community”.(P5, FGD, Male non-treated Bonsukrom)

### 3.6. Impact of the LF Infection on the Community

Study participants said the impact of LF on communities was very severe. The disease had negative impacts on the individuals who have the clinical disease and their family. According to participants, people with the condition are unable to work effectively, which negatively affect their financial status as well as that of their family. The following representative narratives affirmed this position:

“People with the disease cannot do many hard works. They can’t walk about freely, meaning they must sit at one place, and this causes financial problem”.(P2, FGD female non-treated, Mpatase)

“Because the affected person cannot work hard as he used to, the family will suffer because his or her ability to earn money has reduced. And as the man sits at one place, his leg will continue to swell”.(P1, FGD female treated, Mpatase)

Participants also intimated that people with the condition sometimes isolate themselves because of the bad odour from their wounds. The following illustrate these points:

“I know someone, his legs swelled and had a very bad odour. It was so bad that he had to carry perfume around. He sprays himself every now and then before he joins his friends to chat. He is always in trousers to cover his legs. It got to a time he began to stay indoors and died eventually, I believe, out of loneliness”.(P5, FGD, Male treated Bonsukrom)

More than half of the 440 people surveyed (n = 432, 98.2% were aware of the yearly MDA, which has taken place in the areas for more than 15 years. A group that received treatment (n = 214, 99.1%) was more aware than those who had no treatment (n = 224, 97.3%). The proportion of respondents who had ever ingested the medicine during MDA was very high in both groups but significantly higher among the treated (98.6%) than among the non-treated group (76.6%), [*p* < 0.001, CI = 95.75–99.55].

Approximately 51% of those who participated in the study have taken MDA at least seven times since it was first introduced in the district. Between one and more doses, the treated group and the non-treated group differ significantly (*p* < 0.001) in the amount of medication they take.

### 3.7. Importance of Mass Drug Administration

Participants acknowledged the importance of MDA in preventing LF, especially the clinical manifestations of elephantiasis and have therefore reported that the main reason for ingesting the MDA drugs was to protect themselves against the disease. Respondents strongly indicated that the MDA intervention has contributed to the reduction of the disease cases in the community. Respondents also intimated that the MDA drugs are also good for worm and lice control in the community. Some participants shared their opinions on the importance of MDA as follows:

“Every year they bring us some drugs to take and, I believe that is what has been protecting us from this disease […] only few people now have the big legs”.(P4, FGD Female treated, Mpatase)

“Yes, I see that it (MDA drugs) has been very useful. When I was about 20 years old, there were a lot of cases of elephantiasis disease in this community. But nowadays because of the annual drug distribution, it has gone down, though some of us has defaulted on taking the drugs many times”.(P2, FGD, Male non-treated Egyambra)

“The benefits are plenty. I don’t usually take medicine, but I always take the MDA drugs because it deworms me […] it has also eliminated hair lice from the hairs of children in the community, my children used to have lice in their hairs but know, I rarely find one in their hairs”.(P5, FGD female treated, Mpatase)

### 3.8. Adverse Reaction to MDA Drug

Slightly more respondents in the non-treated group (n = 63, 28.9%) than in the treated group (n = 42, 19.5%) reported experiencing adverse effects from MDA drugs. There was a statistically significant difference in the drug reaction between the two groups (Chi^2^(1) = 5.1673; *p* = 0.023).

Nearly one-third of the 105 individuals in both groups who had adverse drug responses, regardless of which group they were in, went to the hospital. General weakness, rashes, swelling, itching, and worm shedding were the five most common side effects that were recorded (Table 6).

The primary reasons given by 74 out of the 105 who reacted and chose not to seek medical attention included “Don’t like hospitals” and “Felt better after few days.” There was no significant difference between the two groups in the reasons given for not seeking hospital care.

Qualitative findings corroborate the quantitative reports regarding adverse reactions of the MDA drug among participants. Some of the adverse reactions reported during qualitative data collection were swelling, rashes, weakness, dizziness, and nausea among other. The following illustrative narratives vividly captured the experiences of the respondents:

“I had a personal experience after taking the drug […] the next day after I took the drug, my leg and wrists began to swell. So, I showed it to the drug distributor, and he gave me a note to send to the hospital […] when I went, I was made to wait for a long time before they attend to me. The following year, I almost refused to take it but when I took it nothing happened to me”.(P7, FGD female treated, Mpatase)

“I personally stop taking the drug because on two occasions when I took it, I had rashes and vomit with blood in my vomitus. I also had swollen face and legs, I was very weak that I could not to anything […] I suffered for about 4 days and when I talked to the CDD, he said to me that they will go by themselves and that I do not need any treatment”.(P4, FGD, Male non-treated, Asemkor)

### 3.9. Adverse Effect and Its Treatment at the Health Facility

Majority (75% out of the 42 treated and 84% of the 65 non-treated participants) who reported having experienced adverse events went to the clinic and said they were given medicine to help with the itch (Piriton), pain killers and ointments in addition to other medications that they did not know. In addition to the medicines that was given to them, 50% out of the 42 treated and 22.6% of the 65 non-treated mentioned that they were given injections (Table 6).

In the opinion of respondents, there is the need to manage people with adverse reactions for free in the health facility. According to them, in the past, MDA drug related adverse reactions were treated for free; however, in recent times, people are being asked to pay for the treatment, especially for medicines, and this is negatively affecting intake of the MDA drugs. The following narrative illustrates this point:

“The days that my brother went to the hospital with his reactions and was treated for free is not the same as we have it today. Today if you go to the hospital with such a problem you will pay. If you do not have health insurance, you will not be attended to. […] you are given a drug, which is making you sick and you must pay for treatment, it is not encouraging, and I think, this is one of the main reasons why some people have stopped taking the drug. I think we must be given a card to present at the health facility when we have reactions from the drug, this will be good”.(P5, FGD, Male treated, Bonsukrom)

### 3.10. Barriers to MDA Coverage and Uptake

Three main sub-themes emerged with regard to barriers to MDA uptake among community members. These include poor planning, CDDs’ lack of commitment, and adverse drug reactions.

Participants in both treated and non-treated groups reported experiencing some physical adverse reactions to the MDA drug at one time or the other, which is making some community members to refuse taking the drug. People are afraid that they may suffer from adverse reactions. Some of the adverse events reported included fever, weakness, swollen legs, and arms. Some participants shared their experiences follows:

“I have stop taking the drugs because I had a bad experience […] I suffered from fever and weakness after taking the drug when I was in school. You know, I was not sick before taking the drug, then suddenly, I felt sick. That is why I stop taking it”.(P3, FGD Male non-treated, Bonsukrom)

“Some complain of feeling sick, especially weakness after taking the medicine, hence do not take it. The men will leave home early to avoid the drug, other will pretend that they have taken alcohol. As for my fellow women, they may even say that they are pregnant busy with other household chores”.(P2, FGD, female treated, Asemkor)

Study participants reported that often they are not given adequate time to get ready for the drugs, insisting that the MDA activities usually suffer from poor planning and coordination with community members to inform them early to enable them to plan their work activities such that they will be at home on the days of distribution. These sentiments are represented in the following narrative:

“One of the reasons why some of us have not been taking the medicine regularly is that we are not inform on time to enable us to stay at home and wait for the distributors […] you will come from the farm or sea and then you are told that the drug distributors came in your absence. Sometimes too the information comes to the community only a day or two before the distribution, by which time you have planned your work activities already […] you must feed your family, so you leave home”.(P6, FGD, Male non-treated, Nyameyekrom)

Some respondents felt that, in some cases, the CDDs are either not committed or motivated to do the work and therefore may not even enter every home in the community. It also emerged that CDDs often complain to community members that the drug distribution work is voluntary, which comes with no pay, so they do not care whether people take the drug or not. These positions were represented in the following narratives:

“I will say that part of the problem is from those who bring the drugs to us […] the volunteers, I can say that they are not committed to the work because they are always complaining that they are not pay for the work, so nobody should worry them […]. At one time I got angry and refused to take the drug, but my son spoke to me to take it because it is good for me”.(P4, FGD, Male treated, Egyambra)

They (CDDs) often just ask of a particular person and when they are told that he or she was not at home, they will not enter that home […] you know that there are other people in that house, but they do not care. I think that they do not like the work […] maybe they must be motivated by paying them for doing the work, I believe when that is done, they will work harder because we all like money (P3, FGD, Female non-treated, Achowa).

### 3.11. MDA Participation among Both Groups

During the over 15 years of MDA administration in the district, only 28% of the non-treated group reported taking the treatment more than 7 times, compared to 69% of the treated group (Figure 1). When asked how many times treatments were taken in the last 5 years, it came to light that 51% of respondents in the non-treated group had never taken the drugs even once in the last 5 years (Figure 2). However, 61% among the treated group took the medication consistently in the past five years, with another 18% of them taking it four times in the past five years. Among the non-treated group, 17% reported they were always busy, 17% reported that they travelled and a considerable 67% reported other reasons for not taking part in the MDA. These include dislike for medicine, not considering the MDA as necessary, religious beliefs, low risk perception, etc. Interestingly, majority of the respondents in both groups (98% among treated and 81% among non-treated) were of the view that MDA is an important intervention that has helped to reduce the number of elephantiasis cases in communities, though this was significantly more among the treated group (*p* < 0.001).

We found that six out of the 244 non-treated individuals have reported that they were not aware of MDA activities. Of those who were aware of MDA activities in the district, 51 reported that they have never taken the MDA drugs. Thus, 57 out of the 224 (25.45%) have never taken any MDA drugs in the district (Figure 3).

### 3.12. Strategies to Increase MDA Coverage

Respondents suggested several strategies to be taken to increase participation in MDA activities. Some of these strategies are the need to provide adequate information on the importance of ingesting the MDA drugs to the people on a regular basis, using local media to create awareness about the MDA activities in communities and informing community members about an impending MDA exercise, which must be done repeatedly for over a week or two. Respondents believe that this will enable community members to plan by rescheduling some activities to make themselves available on the day of MDA. These views were summed up in the following narratives:

“The dates and times should be communicated to us earlier just like the nurses do for weighing (child welfare clinic). In most cases, you will be in the house and the drug distributor will come and say that the time for you to take the drug has come. […] the timing should be communicated to us at least a week before they come”.(P5, FGD, Male treated, Bonsukrom)

“It will be good to make announcement using the local FM stations, whenever the drugs are to be distributed […] this will help us to plan and make ourselves available on the day that they will bring the drug to us”.(P3, FGD female non-treated, Mpatase)

“I believe that there is a need for sensitization and information to be done first to make sure that people are well informed about the need for the drugs […]. They understand why everybody should take the drug, also they need to stress the point that when you take the drug and have problems (adverse reaction) you will be taken care of for free at the health facility. I believe that this will encourage people to take the drugs”.(P2, FGD, Male treated, Nyameyekrom)

When respondents were asked to share their opinions on the E&T strategy, they believed that some of the non-treated persons will accept to take the drug but that will also depend on who and how they are engaged. Some are of the view that, if the same CDDs will be involved, then they must be trained to know how to talk to the people as well have time for those they will be engaging in dialogue. The following narratives captured these views:

“I believe that if the engagement is done well by having time for them to explain the importance of the MDA drugs to them, majority of them will take it […] those doing the dialogue must be prepared to visit even in the night like between 7 and 9 pm, so that they can meet them, especially the men at home”.(P4, FGD, Female treated, Asemkor)

“That will be a very good thing to do for us […] I will take the drug now because I now understand the importance of the drug to be useful for preventing infection and I know that many people like me will take it if you come to us at the right time when we are at home, like today”.(P3, FGD, Male non-treated, Mpatase)

Regarding the T&T strategy, respondents indicated that some people would take the medicine only when they are sure that they have the infection in their blood, especially those who have some education. For this group of people, testing will be important for them because they will not take the medicine unless they know that they have the disease. Others will accept to be tested just to be sure that they do not have the disease in their blood, though they may take the medicine even if they do not have the disease. The following narratives captured these opinions vividly:

“I believe that some people would like to test to be sure that they have the disease before they will accept to take the drug, especially those young men who have small education and think that they know everything”.(P2, FGD, Male treated Egyambra)

“I am sure that people like me (non-treated) will like to be tested before taking the treatment […], this is a good thing because I will also know whether I have the disease in my blood or not […], I do not think that anybody will refuse the treatment when they know that they have the disease in their blood”.(F4, FGD, Male non-treated, Bonsukrom)

### 3.13. Influencing Non-Treated to Take MDA Medication

Similar views were shared by both treated and on-treated participants when they were asked “Who can influence those who do not usually take the MDA drugs to do so regularly?”. It came to light that the chief of every community is the most influential significant leader in the lives of community members because chiefs have the power to sanction anyone who refuses their instructions and, in extreme cases, can even banish people from the community, either for life or for a period. Mentions were also made of parents, spouses, teachers, religious leaders, and friends. Participants from both groups, however, maintained that, in practice, the immediate person to influence people to take the drugs varied from household to household since the influence of each person listed above varied in each household. A respondent from the treated group was of the view that, should there be sanctions for those who refused to take the drugs, then the chief will be the most influential person that can make people take the drug because he will have the authority of the community behind him to punish defaulters. These views were represented in the following responses:

“I believe that the chief is the one that everybody listens to, when he speaks, so if he talks to those of us who refused to take the drugs, I believe we shall obey him […] I must add that parents and husbands and sometimes teachers can influence people to take the drug […] pastors or imams can also do same but I must add that for grownups like me, we have the responsibility for ourselves”.(P3, FGD, Female non-treated, Achowa)

“I think that the chief can only influence people to take the drugs when they put in place some forms of punishment for those who do not take the drug […], you see, the chief can only influence people by enforcing the rule that has been put in place to punish defaulters, especially those that intentionally refused to take the drug […]. I believe that parents can influence their younger children to take the drug, none of my children can refuse to take the drug when I have taken it myself”.(P5, FGD, Male treated, Egyambra)

## 4. Discussion

This study is part of the efforts to implement strategies aimed at addressing some of the challenges to MDA. This has become important after implementing over 16 rounds of MDA without breaking transmission in some districts, commonly referred to as ‘hotspots’ [5,12]. It has been established in an earlier study that some people in this district claimed they have either never taken the MDA drugs or have stopped taking it for fear of side effects [5], which confirmed what was reported from other studies [23,24]. It has also been established that not participating in MDA was significantly associated with LF infection, thus those who systematically avoided taking the intervention drugs were more likely to have microfilariae infections [22], which may serve as a reservoir to sustain transmission. The process of engagement placed emphasis on frank and open discussion of the issue of side effects of the MDA drugs and reassuring participants that most of the side effects may not require medical interventions and, if they do, they could access health care from the nearest health facility for free.

Disease control and elimination programmes in developing countries must be adapted to be compatible with not only traditional perceptions of aetiology, transmission, treatment, and prevention but also fit well into the cultural, economic, and social lives of the people [7]. This is important to ensure that community members are available to benefit from whatever interventions that are put in place to control or eliminate diseases like LF. As was reported in the study, respondents (both treated and non-treated) were aware of the clinical manifestations of lymphatic filariasis and had local names/terms for them. Although the hygiene/dirty environment causal theory persists and was the most reported causes of LF among respondents, the mosquito causal theory has begun to filter through and was reported by more than 10% of respondents. This should serve as an entry point to propagate the mosquito bite prevention measures to reduce exposure to infective bites in endemic communities.

It is encouraging to learn that the majority of respondents in both groups had mentioned the MDA drug as the most important way of controlling and preventing LF. This knowledge among the people must be capitalized on by health promoters and educators to promote and increase the acceptance of MDA among the people, especially among the non-treated group. This implies that the introduction of MDA as a control/elimination intervention has helped to change the perception among the people that LF cannot be completely treated or prevented, a belief which had led to dependence on practices such as scarification, which has been identified as a risk factor for rapid progression of filarial elephantiasis [7,23]. The use of herbal concoctions, and all kinds of ointments [6,7], had dwindled in the study communities.

Findings that the implementation of MDA in the district has led to the reduction in perceived risk of infection among the people must be managed in a way that will not affect the MDA uptake in the district, which has remained one of the hotspot districts in Ghana [5,20]. As much as possible, the implementation of MDA, which has led to the perception that the disease has gone down in the communities, must be used in promotional messages by encouraging the people to make sure that everybody in the district takes the MDA drugs to enable a complete elimination of the disease from the communities once and for all.

The impact of LF related clinical manifestations (elephantiasis and hydrocele) was considered very serious, which affects the economic life and livelihood of the infected and affected persons. This confirms what was reported in earlier studies from the district [6,7]. Findings also confirmed earlier reports [6,7,25,26] that patients with visible clinical manifestation, such as big hydrocele and elephantiasis, suffer discriminations in the form of teasing and making some of the affected persons to isolate themselves from social and public engagements. Efforts should be directed at stigma reduction strategies such as making people aware that anybody can be affected by the disease, stressing the fact that everybody living in the district is at risk of infection and that people are infected through no fault of theirs. Assurance should also be given to the public that one cannot get the disease by associating with those affected. In addition, the affected should be supported to rebuild their self-confidence by helping them to engage in livelihood activities that will reduce their dependence on other people for sustenance. It is important to state that, as we work on stigma reduction in the community, we must not shy away from drawing people’s attention to the pains that the disease inflicts on the patient and his/her family members compared to the pains of adverse reactions after taking the MDA drugs, which will help to eliminate the disease completely such that no one will suffer from the pains caused by the disease in future. This must be communicated with pictures to achieve the desired impact, especially on the non-treated persons in the district.

To derive the maximum benefits from the ‘E&T and T&T’ interventions targeting MDA non-treated persons will require the retraining of the CDDs to equip them with skills for building rapport, ability to treat people with respect and have time for them when engaging them. The CDDs must also be equipped with current educational posters and manuals to guide them in their engagement efforts to educate the non-treated on the need for everyone to take the medication as the only way of protecting themselves and their loved ones now and in the future. This is important as both treated and non-treated groups were of the view that the success or failure of the intervention strategy will depend on the attitudes of the CDDs toward the non-treated persons.

It is worthy of note that respondents, both non-treated and treated, were of the view that the ‘E&T’ or ‘T&T’ strategies will be acceptable to the non-treated persons in the district. Thus, those who refused to take the medicine because they did not perceive themselves to be at risk of infection will accept to be tested and, when positive, they will take the medicine. Should the implementation of that strategy be successful, it will open a new window of opportunity to explore further the efforts to break transmission in hotspot districts. These strategies are currently under evaluation.

The recognition of the chief in each community as the most significant influential person whose involvement in the MDA promotional activities could lead to non-treated persons to ingest the MDA drugs must be explored. This position is tenable because, in the cultural context, chiefs have power to sanction anyone who disobey their instructions, and this is in line with what was reported in a malaria study, where the involvement of the chiefs and respect for community social structure and protocols have led to an overwhelming acceptance and participation in an intermittent preventive treatment of children [27]. Nonetheless, the socio-cultural context on rewards and sanctions must be taken into consideration.

## 5. Conclusions and Recommendations

This study was conducted as part of the efforts to implement new strategies to improve MDA coverage, especially when it is established that some people in the study district have either never taken the MDA drugs or have stopped taking it mainly for fear of side effects and low risk perceptions. Respondents were aware of the clinical manifestations of LF. The hygiene/dirty environment causal theory still dominate the reported causes among respondents and will need to be addressed through effective communication and education. The good news is that the mosquito causal theory has begun to filter through and was reported by more than 10% of respondents and health promotional messages must take advantage of this. The MDA drug was reported as the most important means of managing and preventing LF infections, which has led to the reduction of clinical cases of LF in the communities as well as reduced perceived risk of infection and vulnerability among the people. The impact of LF related clinical manifestations (elephantiasis and hydrocele) on the economic life and livelihood of the people, and stigmatization of the affected persons were recognized. Respondents believe that the ‘E&T and T&T’ interventions targeting MDA non-treated may be acceptable to the people, especially when the CDDs are retrained on how to relate with non-treated persons. The need to involve the chiefs in MDA promotional activities should not be overlooked because they have the power to influence people to participate in MDA activities.

Based on the study, a poster was developed (in English and the dominant local languages) with the main LF messages to improve knowledge of the disease and response to MDA. The following recommendations are made in Table 7.

## Figures and Tables

**Figure 1 tropicalmed-07-00273-f001:**
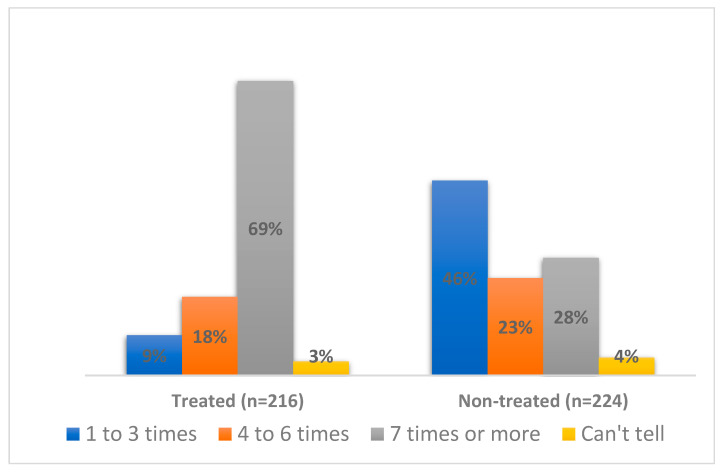
Percentage distribution of times respondents reported taking the MDA drugs.

**Figure 2 tropicalmed-07-00273-f002:**
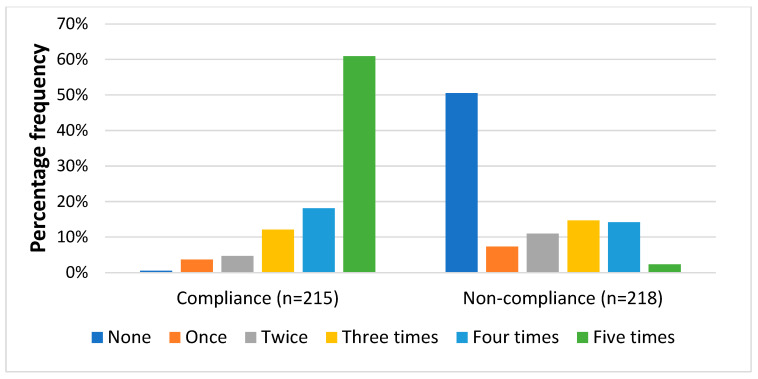
Number of times respondents reported taking the MDA drugs during the last five MDA rounds.

**Figure 3 tropicalmed-07-00273-f003:**
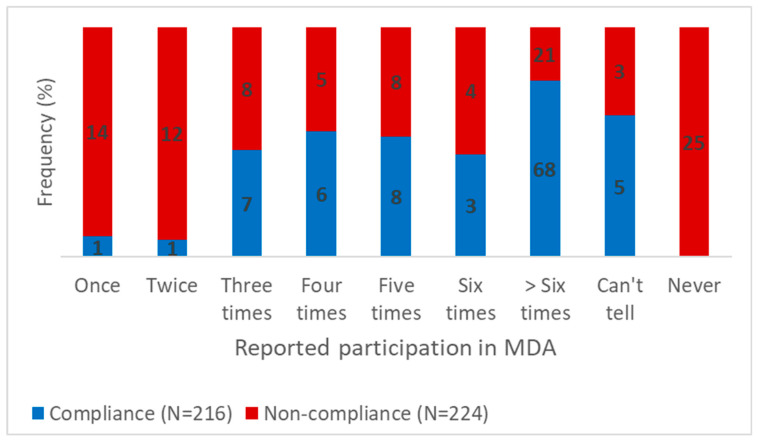
Number of times respondents reported taking the MDA drugs since its introduction in the district.

**Table 1 tropicalmed-07-00273-t001:** Socio-demographic characteristics of survey participants.

Variable	Treated n (%)	Non-Treated n (%)	Total n (%)	Chi Square *p*-Value
**Gender**				0.200
Male	91 (42.1%)	108 (48.2%)	199 (45.2%)	
Female	125 (57.9)	116 (51.8%)	241 (54.8%)	
**Age group**				<0.001
18–34	68 (31.5%)	112 (50.0%)	180 (40.9%)	
35–64	115 (53.2)	98 (43.8%)	213 (48.4%)	
65+	33 (15.3%)	14 (6.2%)	47 (10.7%)	
**Level of education**				0.630
No education	56 (25.9%)	49 (21.9%)	105 (23.9%)	
Primary	47 (21.8%)	59 (26.3%)	106 (24.1%)	
Junior High School or above	112 (51.9%)	115 (51.3%)	227 (51.6%)	
Others	1 (0.5%)	1 (0.4%)	2 (0.5%)	
**Employment Status**				0.350
Employed	194 (89.8%)	191 (86.4%)	385 (88.1%)	
Unemployed	15 (6.9%)	24 (10.9%)	39 (8.9%)	
Others (retired, student)	7 (3.2%)	6 (2.7%)	13 (3.0%)	

**Table 2 tropicalmed-07-00273-t002:** Socio-demographic characteristic of qualitative study participants.

Community	Type of Participants		Data Collection Strategy	Sex	Number of Participants
Achonwa	4 Treated	3 non-treated	FGD	Females	7
Asemkor	5 Treated	2 non-treated	FGD	Females	7
Bonsukrom	4 Treated	2 non-treated	FGD	Males	6
Egyambra	5 Treated	2 non-treated	FGD	Males	7
Mpatase	5 Treated	2 non-treated	FGD	Females	7
Nyameyekrom	5 Treated	3 non-treated	FGD	Males	8

**Table 3 tropicalmed-07-00273-t003:** Causes of Lymphatic Filariasis disease reported *.

	Treated (n = 216)	Non-Treated (n = 224)	Total
Causes Reported *	n (%)	n (%)	n (%)
Poor hygiene/Dirty environment	157 (72.7)	181 (80.8)	338 (76.8)
Don’t know	59 (26.6)	43 (19.2)	102 (23.2)
Mosquitoes	34 (15.7)	26 (11.6)	60 (13.6)
Worms	10 (4.6)	12 (5.4)	22 (5.0)
Rain/Standing water	7 (3.2)	11 (4.9)	18 (4.1)
Drinking from streams or rivers	11 (5.1)	7 (3.1)	18 (4.1)
Walking barefoot	11 (5.1)	5 (2.2)	16 (3.6)
Long contact with sea water	7 (3.2)	3 (1.3)	10 (2.3)
Pollution	5 (2.3)	2 (0.9)	7 (1.6)
Hereditary	6 (2.8)	1 (0.5)	7 (1.6)
Witchcraft/Curse/Spiritual	16 (7.4)	17 (7.6)	6 (1.4)
Oily foods/peanuts		5 (2.2)	5 (1.1)
Bad wind		2 (0.9)	2 (0.5)
Sweet foods/sugar		1 (0.5)	1 (0.2)

* Multiple responses allowed.

**Table 4 tropicalmed-07-00273-t004:** Signs and symptoms of LF disease.

	Treated (n= 216)	Non-Treated (n = 224)	Chi2/*p* *
	Frequency(%)	Frequency (%)	
Swollen legs	209 (96.8)	206 (91.9)	4.718/0.209
Sores on the body	13 (6.0)	18 (8.0)	0.683/1.000
Don’t know	6 (2.8)	15 (6.7)	3.715/0.377
Swollen scrotum (hydrocele)	4 (1.9)	3 (1.3)	0.185/1.000
Swollen arms and hands	4 (1.8)	3 (1.3)	0.185/1.000
Others	2 (0.9)	3 (1.3)	0.167/1.000
High body temperature (fever)	0 (0.0)	3 (1.3)	2.913/0.703
Swollen breasts	0 (0.0)	2 (0.9)	1.937/1.000

Multiple responses allowed., * Pearson chi2(1)/Bonferroni-adjusted *p*-values.

**Table 5 tropicalmed-07-00273-t005:** Means of control and prevention of LF *.

	Treated (n = 175)	Non-Treated (n = 162)	Chi2/*p* *
Means to control or prevent	Frequency (%)	Frequency (%)	
Mass Drug Administration (MDA)	148 (84.6)	124 (76.5)	3.483/0.372
Keeping clean environment	11 (6.3)	19 (11.7)	3.073/0.478
Don’t know	13 (7.4)	17 (10.5)	0.975/1.000
Use/Drink clean water	2 (1.1)	7 (4.3)	3.269/0.424
Bednet/Insecticide spray/Mosquito coil usage	8 (4.6)	5 (3.1)	0.500/1.000
Others	3 (1.7)	5 (3.1)	0.683/1.000

* Multiple responses allowed.

**Table 6 tropicalmed-07-00273-t006:** Reported adverse reactions to MDA drugs and measures taken.

	Treated	Non-Treated	Total	Chi^2^/*p* *
	n (%)	n (%)	n (%)	
**6.0 Ever React to Drugs (n = 433)**				
Yes	42 (19.5%)	63 (28.9%)	105(24.3%)	5.1673/0.023
No	173 (80.5%)	155 (71.1%)	328(75.7)	
**6.1 Reaction to drugs # (n = 105)**				
Weakness	6 (14.3%)	15 (23.8%)	21 (20.0%)	1.429/1.000
Rashes	8 (19.1%)	11 (17.5%)	19 (18.1%)	0.043/1.000
Swelling	8 (19.1%)	10 (15.9%)	18 (17.4%)	0.179/1.000
Itching	9 (21.4%)	8 (12.7%)	17 (16.2%)	1.415/1.000
Worm discharge	7 (16.7%)	10 (15.9%)	17 (16.2%)	0.012/1.000
Nausea	2 (4.8%)	8 (12.7%)	10 (9.5%)	1.842/1.000
Dizziness	3 (7.1%)	7 (11.1%)	10 (9.5%)	0.461/1.000
stomach-ache	1 (2.4%)	4 (6.4%)	5 (4.8%)	0.875/1.000
Severe headache	0 (00%)	3 (4.8%)	3 (2.9%)	2.059/1.000
Others	3 (7.1%)	14 (22.2%)	17 (16.2%)	4.223/0.399
**6.2 Visit to health facility (n = 105)**				0.0305/0.861
Yes	12 (29.0%)	19 (30.0%)	31(29.5)	
No	30 (71.0%)	44 (70.0%)	74 (70.5)	0.0
**6.2.1 If no, why didn’t you go to the health facility to report the incident? (n = 74)**				10.6149/0.060
I know that health workers will not pay attention to me	1 (3%)	0 (0%)	1 (1.4)	
I know the health facility has no medicine for adverse reaction	0 (0%)	1 (2%)	1 (1.4)	
High cost of treatment at health facility	0 (0%)	4 (9%)	4 (5.4)	
Don’t like hospitals	4 (13%)	15 (34%)	19 (25.7)	
Felt better after some days	15 (50%)	12 (27%)	27 (36.5)	
Others (specify):	10 (33%)	12 (27%)	22 (29.7)	
**6.2.2 If yes, treatment given at health facility (n = 31)**				233.7281/0.000
Given medicine which I don’t know it name	9 (75.0%)	16 (84.2%)	25 (80.7%)	
Injection	7 (50.0%)	1 (5.3%)	7 (22.6%)	
Given Piriton to stop the itching	1 (8.3%)	1 (5.3%)	2 (6.5%)	
Given pain killers for headache	0 (0.0%)	1 (5.3%)	1 (3.2%)	
Given ointment for swelling	0 (0.0%)	1 (5.3%)	1 (3.2%)	
Others	1 (8.3%)	1 (5.3%)	2 (6.5%)	

# Multiple choices allowed. * Pearson chi2(1)/Bonferroni-adjusted *p*-values.

**Table 7 tropicalmed-07-00273-t007:** Recommendations to issues identified.

Issue Identified	Recommendations
Education	Education needs to be improved for the health workers, CDDs and for the communities to address the misconceptions surrounding the transmission of the disease.
Information on MDA	Providing adequate information on the importance of MDA through information, education and communication (IEC) materials could help increase intake of the drugs. The information should include the fact that most, if not all, medicines are likely to activate some reactions in some people, which is part of the natural characteristics of medicine.
Timing of MDA	More time needs to be given for the MDA or more appropriate time identified to enable community members participate in the process.
Response to adverse events	The recommended protocols for addressing adverse events to treatment need to be enforced, as respondents complained of being asked to pay for treatment during adverse events.
Number of non-treated	An important percentage of non-treated exists in the district, with reasons not related to the absence during MDA. These individuals will need to be targeted to assess the level of infection in this population.

## Data Availability

All data (qualitative and quantitative) generated and analysed during this study are included in this paper. Raw qualitative data are not available and will not be shared, as this would compromise the protection of participants’ identity; however, some data could be made available upon a reasonable request from the corresponding author, cahorlu@noguchi.ug.edu.gh.

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
