# Peer review of "A Comparative Study of Lymphatic Filariasis-Related Perceptions among Treated and Non-Treated Individuals in the Ahanta West Municipality of Ghana"

_tropicalmed, 2022, doi:10.3390/tropicalmed7100273_

Round 1

Reviewer 1 Report

Well-conducted mixed methods study. Authors can discuss issues regarding adverse events more. T&T strategy is less discussed in the manuscript. Do authors feel E&T is superior to T&T?

Reviewer 2 Report

The objectives of this mixed (qualitative/quantitative) study are clear and interesting: to test whether there are significant differences in knowledge, perception and behaviour between people who agree to participate in mass treatment for lymphatic filariasis and those who do not. The issue is relevant because among the latter there could be a variable number of infected individuals who would remain as a potential source of transmission.

However, the paper has some major problems that compromise its quality and make it difficult to draw useful conclusions.

Main observations

1.         I am not an expert in qualitative studies, but I absolutely recognize their importance and necessity. However, the quantitative part needs to be rigorous. It is not at all clear how the sample was selected, since the treated were chosen randomly from the register (no objection), while the untreated 'were chosen on purpose'. What does this mean? The authors should explain well and in detail how the recruitment of the untreated took place.

2.         Still referring to the quantitative part: by what criteria was the number of subjects to be interviewed chosen? (sample size)

3.         Unfortunately, recruitment problems may heavily influence the results. For example, in Table 1, I find a highly significant difference in the age distribution of the two groups, but if I do not know the selection criteria, this tells me nothing, and I rather think of a selection bias.

4.         This is also reflected in the results for the main variables researched, for which, in most cases, no significant differences are found between the two groups. Unfortunately, a possible selection bias may have made it impossible to detect significant differences actually present, if any.

5.         Considering all this, in the end the main problem with this study is that it does not in fact find any substantial differences between the two groups, and thus does not allow us to point to evidence-based strategies for trying to change certain knowledge, attitudes and behaviour to achieve better compliance in the untreated group. Unfortunately, we do not know whether this lack of differences is real or artefactual.

In conclusion, the study is well done from a qualitative point of view and details many of the responses received from participants regarding infection, its aetiology and transmission, treatment and side effects. However, due to the quantitative problems I mentioned earlier, it does not suggest any truly evidence-based direction of activity.

Reviewer 3 Report

In the manuscript entitled “A comparative study of lymphatic filariasis-related perceptions among treated and non-treated individuals in the Ahanta West municipality of Ghana” the authors discuss the detailed comparative analysis of the perceptions of the treated vs untreated persons in the Ahanta West municipality of Ghana. Below are the suggestions to improve the manuscript.

1.    Lines 41-44: Respondents in both groups believed that, when engaged properly, most non-treated persons will accept to take the drug but insisted that community drug distributors (CDDs) must be trained to effectively engage people and have time for those who will be engaging in dialogue.  [Sentence structuring and grammar should be corrected throughout the manuscript].

2.    Lines 49-50: Respondents called for retraining of the CDDs to enable them to execute their roles in the MDA implementation more effectively. [Sentence structuring and grammar should be corrected throughout the manuscript].

3.    Proper Key words should be selected. Why the words ‘and treat’ are used twice in the Keywords list?

4.    Lines 85-86: Prior to, 85 21]. What is the meaning of this sentence?

5.    Lines 90-92: In this study, we defined the “non-treated” to include those who missed three consecutive MDAS or those who missed 3 out of 5 consecutive MDAs or those who had never taken MDA medicine.

6.    Where the study participants age, sex, educational background, social, vocational background/s matched? How are these factors influencing the outcome of this study? The authors should discuss this point.

7.    A. Lines 138-140: The In-depth interview (IDI) was conducted in the Twi language by an interviewer, who was supported by a note taker. The IDI sessions lasted about 30-40 minutes.

B.    What is the expanded form of IEC? [The abbreviations must be expanded the first time and this pattern should be used throughout the manuscript].

8.    Lines 146-150: FGD participants were intentionally selected. They included adult community members, aged 18 years and above, who have lived in the community for not less than 10 years and have either participated or not participated in the MDA prior to data collection. Community drug distributors (CDDs) were also involved for each FGD session. [Sentence structuring and grammar should be corrected throughout the manuscript].

9.    What is the role of corruption? Sincere efforts should be made to reduce/mitigate corruption among the higher officers who play a decisive role/s in the implementation of MDA program. The authors should discuss this issue.

Round 2

Reviewer 2 Report

I thank the authors for providing a revised version. However, I am afraid that my remarks were only very partially addressed and the sampling problem persists, with all consequences that I have already highlighted on the main findings and conclusions.